

# Mapping Sea Ice Concentration in the Canadian Arctic with CryoSat-2

Amy E. Swiggs[1], Isobel R. Lawrence[2], Andrew Shepherd[1]

[1] Centre for Polar Observation and Modelling, Northumbria University, Newcastle Upon Tyne, NE1 8ST, United
Kingdom
[2] European Space Agency, ESRIN, Frascati, 00044, Italy

*Correspondence to*: Amy E. Swiggs (amy.swiggs@northumbria.ac.uk)

**Abstract.**

Sea ice concentration (SIC) is an essential parameter for understanding environmental change in the polar regions. Historically, SIC has been determined using satellite passive microwave (PMV) radiometry, and this has revealed a progressive decline in the extent of the ice cover in the Arctic since records began in 1979. At regional and local scale, classifications based on satellite radar and optical imagery are practical. Here, we use CryoSat-2 to derive a new SIC product in the Canadian Arctic (CA), a region that is vital for shipping, freshwater production, and multi-year ice transport but is frequently excluded from pan-Arctic sea ice satellite observations. The 300 m along-track sampling of CryoSat-2 allows the fine-scale distribution of sea ice to be resolved, and an empirical correction for the overestimation of leads and misclassification of floes allows SIC to be determined. In general, spatial and temporal variations in SIC determined from CryoSat-2 are in close agreement with those determined from PMV and synthetic aperture radar (SAR) imagery in ice charts. Across the CA region, the root mean square difference (RMSD) between SIC determined monthly from CryoSat-2 and PMV and weekly from ice charts are 8.4 and 10 %, respectively. A local comparison to SIC determined from 82 cloud-free Landsat 8 scenes acquired in the central CA shows an RMSD of 3.3%. Our findings highlight the complementarity of SIC records determined from CryoSat-2 and their potential to expand our knowledge of ice conditions in the CA.

## 1 Introduction

Arctic and Antarctic sea ice concentration (SIC), defined as the fraction of the ocean covered by sea ice, has been observed using passive microwave (PMV) radiometry sensors onboard satellites since the 1970s (Cavalieri et al., 1984; Comiso, 1986). These observations have shown declines in SIC and sea ice extent (SIE) across the Arctic since 1979, with record losses in 2007, 2012, and 2020 (Meier and Stroeve, 2022). This is intersected by strong interannual and spatial variability; for example, dramatic losses of ice in the Beaufort Sea and Baffin Bay have not been matched in the Canadian Arctic Archipelago (CAA) (Glissenaar et al., 2023), and unlike the rest of the Arctic, the Bering Sea exhibited positive winter SIE trends from 1979-2016 (Onarheim et al., 2018). High resolution, regional observations of sea ice are therefore essential for understanding climate processes and trends in the Arctic.

Despite its importance for freshwater and multi-year ice transport (Beszczynska-Möller et al., 2011; Melling, 2022), sea ice-dependent species (Lange et al., 2019), and trans-Arctic shipping (Dawson et al., 2018), the Canadian Arctic (CA) is frequently excluded from pan-Arctic sea ice satellite observations (e.g., Landy et al., 2022; Tschudi et al., 2020; Kacimi and Kwok, 2022) due to the challenges of satellite retrievals and in-situ



measurements. Shipping has increased in the CA since the early 2000s (Pizzolato et al., 2014; Dawson et al., 2018), but the Archipelago's geometry means thick, hazardous sea ice builds up in the narrow channels, generating seasonal landfast sea ice that is a significant hazard to ship navigation (Howell et al., 2013). This risk is estimated

to be decreasing as the percentage of old, thick ice in the CAA decreases (Melling et al., 2022). However, this is complicated by recent ice flux events that indicate warming Arctic temperatures are increasing multi-year ice (MYI) transport into the CAA, continuing to make it a highly hazardous shipping route (Howell et al., 2019; Howell et al., 2024). High-resolution observations of sea ice in the CA are therefore of vital importance.
PMV sensors offer fine temporal resolution, generally on daily timescales, but relatively coarse spatial resolution,

often of 25 km x 25 km (e.g., Cavalieri et al., 1984; Comiso 1986; Tonboe et al., 2016) and can therefore struggle to capture complex fine-scale ice dynamics. Recent advances have also been made using satellite synthetic aperture radar (SAR) due to its high spatial sampling and ability to monitor through clouds. However, due to the demanding processing capacity required, and the limited availability of uncertainty quantification in the retrieval methodologies (Wulf et al., 2024), these methods have previously been limited to regional or operational

forecasting. In the CA, ice charts produced by the Canadian Ice Service (CIS) have been vital for monitoring sea ice changes on long timescales, particularly as observations from PMV sensors in the CA have been found to vary in performance (Agnew and Howell, 2003; Wulf et al., 2024). However, data from the CIS can lack spatial variability, as polygon concentrations are rounded to the nearest tenth (Canadian Ice Service, 2009). Satellite altimetry thus presents an opportunity for utilising high resolution point measurements to produce an alternative,

complementary estimate of SIC in the CA. Satellite altimetry has been successfully used for estimating Arctic-wide sea ice thickness (SIT), and while previous work has highlighted the use of laser altimetry in estimating SIC (Liu et al., 2022), radar altimetry has been underutilised despite its greater temporal sampling due to not being hampered by clouds. The radar altimeter CryoSat-2 (CS2), launched in 2010, provides the longest data record from a satellite altimeter. CS2 has a pulse-limited footprint of approximately 0.31 km by 1.67 km along- and

across-track (Wingham et al., 2006) and can therefore resolve ice pack concentration to a high spatial resolution. Here, we present a SIC estimate from CS2 in the CA from 2010 to 2023. We employ a bias adjustment generated from Landsat 8 (LS8) validation imagery to account for the radar altimetry challenges of off-nadir ranging and misclassification of floe waveform returns (Swiggs et al., 2024). With this adjustment, we generate a CS2 SIC product from October to May in the CA and undertake a comprehensive comparison to existing SIC estimates

from the NASA Team (NT) algorithm and CIS ice charts. In addition, we utilise the high spatial resolution of LS8 imagery to provide a further focused comparison in areas of different ice types and climatic conditions. Finally, we consider the impact of spatial and temporal sampling on satellite SIC estimates, treatment of refrozen leads, and discuss the potential for an Arctic-wide SIC product from CS2, exploring the benefits and limitations of radar altimetry for estimating SIC.

**2 Data**

**2.1 Radar Altimetry (CryoSat-2)**

We use CS2 along-track data acquired over the CA to map October to May SIC from 2010 to 2023. CS2 operates in the Ku-band (13.6 GHz) up to 88˚, with a 30-day sub-cycle and a ground footprint of approximately 0.3 km 1.5 km along- and across-track (Wingham et al., 2006). The operational mode depends on the observed surface, and



although CS2 generally operates in synthetic aperture radar (SAR) mode over sea ice, it is operated in SAR interferometric mode (SARIn) over the CA (Wingham et al., 2006) due to its coastal geography. This study therefore uses Level 1b (L1b) SARIn data, available for download from ESA via FTP (ftp://science-pds.cryosat.esa.int/). We apply a bias adjustment to the CS2 waveform returns, based on the results from Swiggs et al. (2024), and further discussed in Section 3. We only use CS2 data from October to May due to the presence

of melt ponds on the surface of Arctic sea ice in summer that make it challenging to distinguish between waveform returns.

### 2.2. Passive Microwave Radiometry (NASA Team)

We use SIC retrievals from PMV to compare to our CS2 estimates. PMV observations of sea ice rely on the variations in emissivity from different surfaces, with different ice and snow types having varying emissivity and thus

brightness temperatures (Cavalieri et al., 1984). Static tie points determined from brightness temperature values over known surface types are used as reference points in the classification. Additional filters are applied to reduce the effects of land spillover and noise from weather and open water roughness. A number of PMV algorithms have been developed for retrieving Arctic SIC, which differ in their usage of frequencies, polarisations, and tie points (Ivanova et al., 2015). We select the NT algorithm (Cavalieri et al., 1984) to compare to our CS2

estimate, as it is one of the main widely used algorithms (e.g., Meier and Stroeve, 2022; Tilling et al., 2018; Lee et al., 2017). Full details of the processing of PMV SIC data in the NT algorithm are outlined in Cavalieri et al. (1997, 1984). The dataset is generated from thermal brightness data from multiple satellite-based sensors, although for the time period in this study, only data from the Special Sensor Microwave Imager/Sounder (SSMIS) is used. The accuracy of NT is estimated to be ± 5 % in winter and up to ± 35 % in summer (Kern et al., 2020).

We download monthly SIC from 2010 to 2023 from the National Snow and Ice Data Center (NSIDC) (https://nsidc.org/data/nsidc-0051/versions/2) which are provided on a 25 x 25 km grid. We regrid the data onto our common 25 x 25 km grid centred on the CA and calculate the SIC as a percentage.

### 2.3. Ice Charts (Canadian Ice Service)

We utilise weekly regional ice charts from the CIS as a further comparison to our CS2 estimates. Since 1996, CIS

ice charts have been compiled by ice analysts, primarily visually examining Radarsat imagery, with additional data sources including aerial, shipping, and operational models (Canadian Ice Service, 2009; Shokr and Markus, 2006). Weekly sea ice information is represented using the World Meteorological Organisation (WMO) egg code, which includes total sea ice concentration, among other parameters, provided for polygons within the CA (Fig. 1). CIS ice charts have been used in numerous studies investigating sea ice in the CA (e.g., Howell et al., 2024;

Tivy et al., 2011; Glissenaar et al., 2023), and we thus utilise them as an additional comparison dataset in our study. Potential systematic errors introduced by data format changes and the introduction of Radarsat imagery as the primary data source (Tivy et al., 2011) are not applicable here as they both were introduced prior to our period of study. Total ice concentration data processed from the Canadian Ice Service WMO egg code format into SIGRID-3 tables are available via FTP from the NSIDC (https://nsidc.org/data/g02171/versions/1) (Canadian Ice

Service, 2009). We use weekly total ice concentration data from 2010 to 2022 in the Western Arctic and Eastern Arctic. At the time of writing, data in the Western Arctic for 2023 were not available for download. We average



the weekly polygon concentrations onto our common 25 x 25 km grid, and find the monthly average percentages of SIC.

**2.4. Optical Imagery (Landsat 8)**

We use 82 LS8 images from 2010 to 2023 that are near-coincident to CS2 along-track retrievals over the CA to provide a final comparison to our CS2 SIC (Fig. 1). These images were used to validate CS2 along-track lead and floe density (Swiggs et al., 2024), and we further use them here to provide an additional spatial and temporal sampling resolution. Launched in 2013 with a 16-day repeat cycle and a swath width of 185 km, LS8 has a spatial resolution between 30 m and 100 m depending on the sensor and can therefore be compared to up to 50 grid cells

from the gridded CS2, NT, and CIS SIC in a single image. The images intersect with CS2 on the same day within a   2-hour   acquisition   window,   available   from   USGS   via   Google   Earth   Engine (https://developers.google.com/earth-engine/datasets/catalog/landsat-8/) (Fig. 1). We map the SIC of each LS8 image using the image brightness temperatures calculated from the Thermal Infrared Sensor. We utilise the temperature anomaly exhibited by leads relative to the surrounding ice floes (Stone and Key, 1993) to segment

the images into leads and floes. We then evaluate the segmentation masks with true colour images derived from the Operational Land Imager sensor Bands 1-7 to ensure selection of the best brightness temperature thresholds. This processing is further outlined in Swiggs et al. (2024) and examples of true colour images with their segmentation masks are shown in Fig. C2. We calculate SIC of the entire image area using the ratio of floes to the total sea surface.


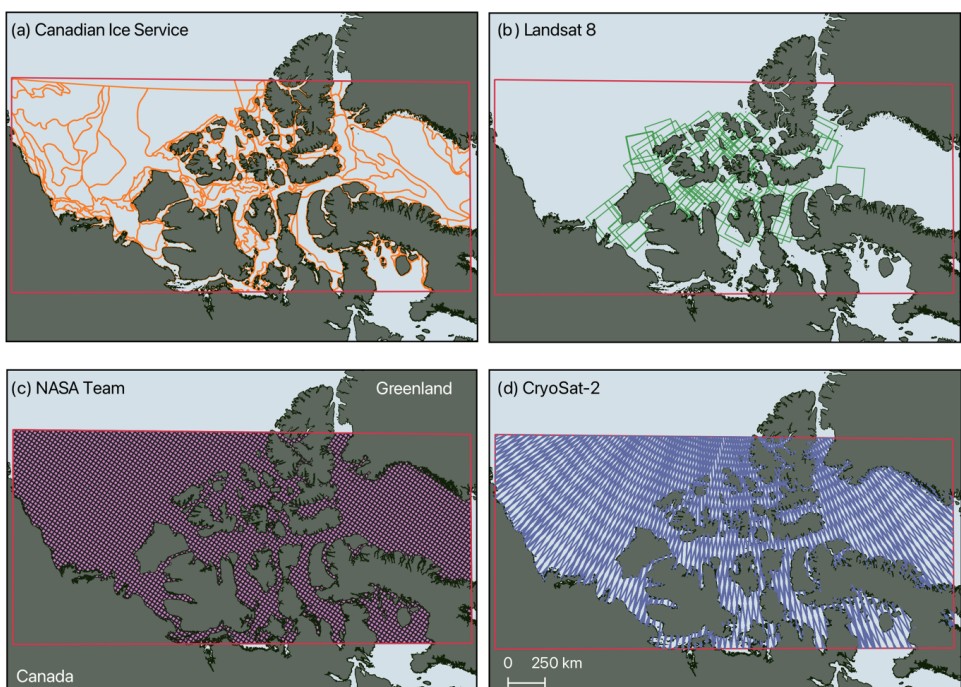



**Figure 1: Summary of the different datatypes used in this study prior to data processing. The red box indicates the study area of the Canadian Arctic, including the Beaufort Sea and Baffin Bay. Each panel shows the original area of the datasets prior to processing (a) The polygons of Canadian Ice Service charts from the Western and Eastern Arctic**

**for an example month (March 2021) (b) Locations of the 82 Landsat 8 images (c) NASA Team on a 25 x 25 km grid (d) CryoSat-2 tracks for an example month (March 2021).**

## 3 Methods

To generate a SIC estimate in the CA, we process along-track CS2 waveform returns into lead, floe, ocean, or ambiguous classifications based on their returned power. Sea ice and the snow that covers it scatters radar pulses,

creating a diffuse echo where the power falls off gradually after the peak, whereas the still water in leads provides a mirror-like surface where the majority of the radar pulse is scattered back to the receiver antenna, returning a specular waveform (Laxon, 1994). This process is complicated by the presence of melt ponds on the surface of Arctic sea ice during summer, which return a specular waveform and can therefore be mistaken for leads (Laxon et al., 2013). We therefore only process CS2 data from October to May. Furthermore, the presence of wind-

generated waves on the open-ocean surface can return a diffuse waveform and therefore be mistaken for an ice floe (Laxon, 1994). To ensure that open ocean is not misidentified as a floe, we follow the methodology of previous authors (Ricker et al., 2014; Tilling et al., 2018; Paul et al., 2018) by employing a threshold to diffuse waveform returns, where the surrounding 25 km grid cell must have a SIC above 75% for the diffuse waveform to be classed as an ice floe.

To classify the returned specular and diffuse waveforms, we measure the ratio of peak power to total power, termed the pulse peakiness (PP), and the backscattering variation for all the waveforms returned from the same footprint location, termed the stack standard deviation (SSD) (Tilling et al., 2018). While there have been different thresholds applied to waveform classification (e.g., Tilling et al., 2018; Ricker et al., 2014; Kurtz et al., 2014; Paul et al., 2018), this was found to have a limited impact on the overall efficacy of CS2 classification in the CAA

(Swiggs et al., 2024). We consider floes to have a PP < 9 and SSD > 4.62 and leads to have a PP > 18 and SSD < 4.62 (Tilling et al., 2018). Waveforms outside of these classification thresholds, caused by complex surface scattering and mixed surface types within the radar footprint (Laxon et al., 2013; Armitage and Davidson, 2014), are discarded as ambiguous.

We apply the bias adjustment generated in Swiggs et al. (2024) to the returned CS2 lead and floe densities. From

a validation using 82 LS8 images in the Northwest Passage, CS2 lead and floe densities were found, on average, to be 14% higher and 45% lower than LS8, respectively. As leads are a highly reflective scattering surface, they dominate the radar backscatter, causing an overestimation of leads within the radar footprint and an underestimation of sea ice in areas of mixed surface types. Furthermore, sea ice floes were frequently identified as ambiguous in waveform classification due to complex waveforms being returned from rough, deformed ice and

snow. To correct for these differences, a bias adjustment to the CS2 lead and floe densities was generated from a simple power law of monthly average lead and floe densities from CS2 and LS8, which we apply to the along-track lead and floe densities from CS2 across the CA.

We bin the CS2 tracks into 25 x 25 km grid cells across the CA, totalling the number of leads, floes, and ocean waveforms identified in each grid cell. We exclude tracks where there are less than 50 valid returns in the grid

cell to ensure the calculated concentration value is representative. Furthermore, if the returned values in the grid



cell are 100% lead, we exclude these grid cells as ambiguous as it suggests that no floes were identified in a 25 x 25 km area, despite the presence of leads that would only occur between floe segments. The high mean distribution of floes, leads, and ocean returns in our SIC calculation shows that a reliable SIC estimate can be produced across the CA (Fig. 2). We estimate the error of our CS2 data to be 5.5% based on the RMSD from the LS8 validation
(Swiggs et al., 2024). We calculate the SIC percentage in each grid cell using the number of floe returns divided by the total number of valid returns (lead, floe, ocean waveforms).

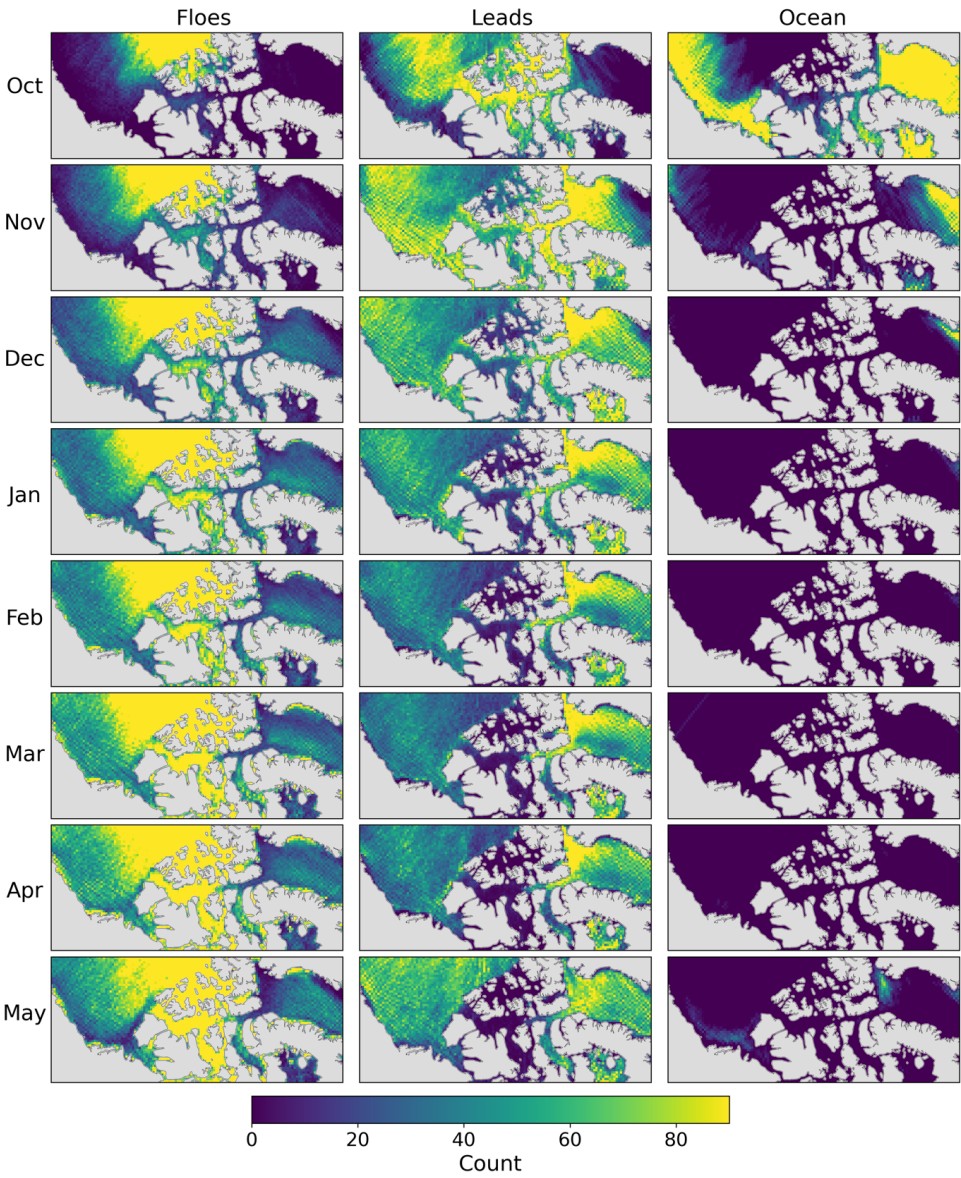

**Figure 2. The mean count of each valid surface type (Floes, Leads, Ocean) within each 25 x 25 km grid cell over the**
**Canadian Arctic, from CryoSat-2 waveform returns.**



## 4 Results and Discussion

### 4.1. Sea Ice Concentration from CryoSat-2

Using CS2, we produce monthly October to May SIC in the CA from 2010 to 2023 (Fig. 3). The CS2 SIC reflects the seasonal cycle of sea ice in the CA, with extremely high SIC throughout Winter and Spring, and low
concentrations and high interannual variability in Autumn following the summer melt season. October SIC, however, remains high in the central and northern channels of the CAA, where a combination of MYI, ice transported from the Arctic Ocean into the channels via the numerous ice flux gates of the Queen Elizabeth Islands, and landfast ice that is fixed to the coastal passages of the CAA are found (Melling et al., 2022; Howell et al., 2016; Howell et al., 2024). The southern Northwest Passages also retain a SIC of above 50% in October, as the
region's geometry preserves high sea ice conditions throughout the year compared to other areas such as Baffin Bay (Howell et al., 2009).

Rapid ice growth occurs between October and January, with mean SIC increasing from 35.7% to 85.7% for CS2, 40.7% to 96.6% for NT, and 52.7% to 95.9% for CIS. The highest concentrations are similarly found in channels of the central CAA, while low to zero ice concentration areas persist in Baffin Bay. In March, the highest mean
SIC and lowest spatial variation in SIC is observed, at $88.6 \pm 5\%$, $97.6 \pm 1.7\%$, and $96.5 \pm 0.7\%$ for CS2, NT, and CIS, respectively. Present in each dataset is an area of low SIC in northern Baffin Bay, which persists and expands in May (Fig. 3), termed the North Water Polynya (e.g., Melling et al., 2001). In the northern CAA, however, an anticyclonic gyre contributes to heavy ridging of sea ice in the Queen Elizabeth Islands that leads to persistently high SIC (Melling, 2002). The CAA icepack from November to July is largely immobile (Howell et al., 2024)
and the region's geometry, landfast sea ice, and the replenishment of thick ice from the Queen Elizabeth Islands somewhat insulate the region from seasonal climatic conditions, reducing its interannual variability (Howell et al., 2013) (Fig. 3b).



Figure caption area with Sea Ice Concentration (%) color bar.





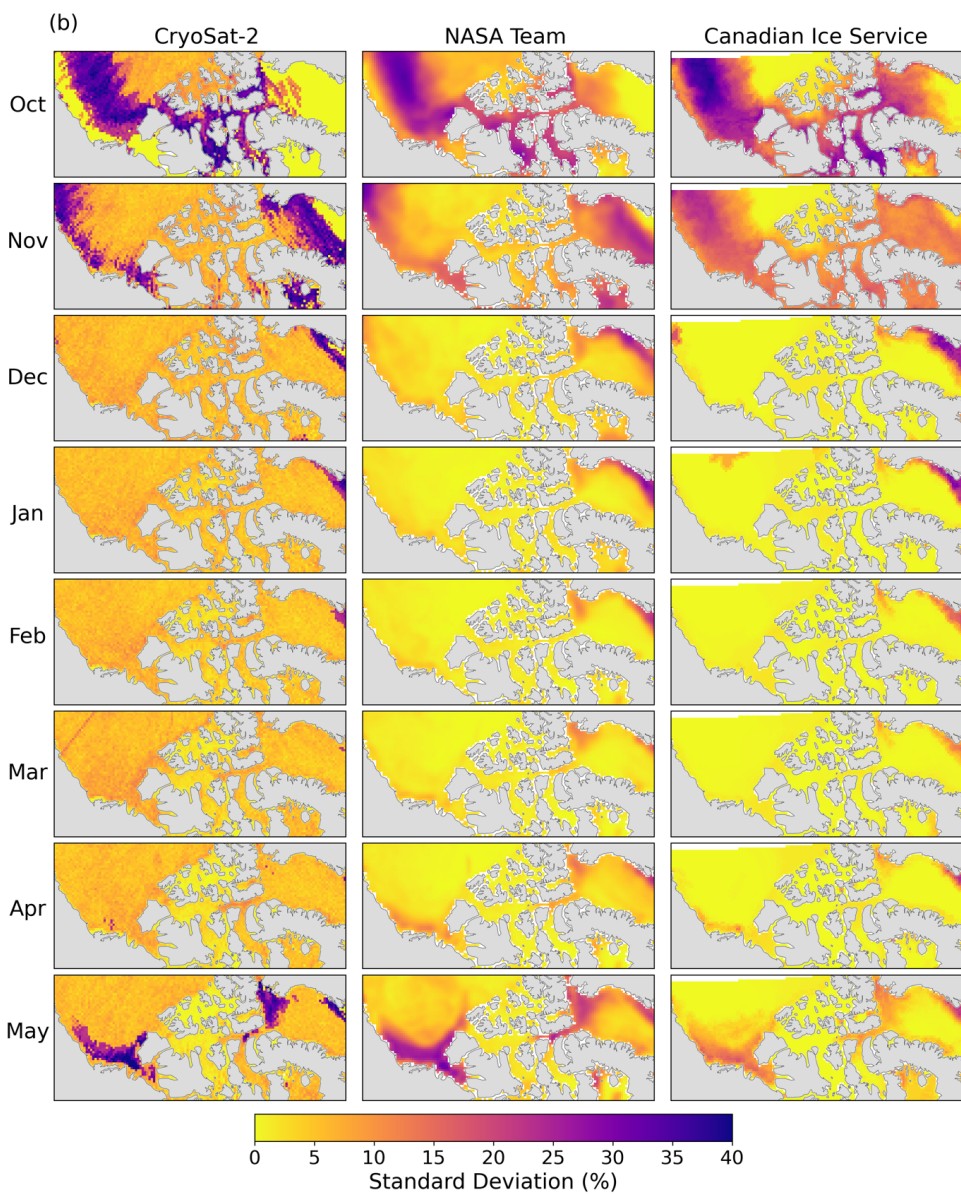


**Figure 3. (a) Mean (2010 to 2023) monthly sea ice concentration maps from CryoSat-2, NASA Team, and the Canadian Ice Service for each season over the Canadian Arctic (b) the interannual variability (standard deviation) of the 2010 to 2023 monthly means.**

## 4.2. Temporal Variations in Sea Ice Concentration

We evaluate our CS2 SIC product by comparing it to estimates determined from other sensors. Monthly SIC from CS2 is highly correlated to NT (r = 0.99) and CIS (r = 0.95) monthly averages (Table A1). The RMSD between CS2 and the other datasets also suggests a good agreement, at 8.4% for NT and 10% for CIS, although there is a



difference between the highest values retrieved by the datasets; NT identifies the highest SIC to be across the Beaufort Sea, while CIS observes the highest SIC to be in the central CAA and Baffin Bay, and CS2 retrieves

generally lower SIC across the study area. Annually (Fig. 4a), the datasets show similar temporal trends, and also exhibit the patterns noted previously, with NT and CIS exhibiting the highest annual SIC. At monthly sampling (Fig. 4b), the expected seasonal cycle of SIC is captured in each dataset, but seasonal differences are present, with the NT algorithm generally recording the highest SIC in spring, while in May and October, CIS retrieves the highest SIC, and in November, the lowest. We consider the causes for these seasonal differences in the following

paragraphs.

The temporal variation in difference between the datasets can be attributed to challenging retrieval conditions. For example, in October, the RMSD between CS2 and CIS is 16.1%, and 11.2% for NT and CIS. The RMSD between NT and CS2 is lower, at 5.7%. Autumn is a challenging season for satellite sea ice retrievals due to the uncertainties introduced by mixed surface types within the sensor footprint, areas of thin ice, and the moving ice

edge (e.g., Armitage and Davison, 2014; Andersen et al., 2007; Andersen et al., 2006). The increased departure between the CIS observations and those from NT and CS2 in October is likely due to a combination of overestimation of SIC by CIS and underestimation of SIC by CS2. Overestimation of SIC in CIS ice charts is likelier to occur in lower ice concentrations, and high variability has been found in the analysis of 'middle' ice concentrations (Cheng et al., 2020; Tivy et al., 2011). For CS2, the application of the SIC threshold to diffuse

waveforms excludes more data in October due to the SIC being below 75% in certain areas (Fig. 2) and could misclassify ice floes in low ice areas (Swiggs et al., 2024).  In addition, higher lead density and thinner ice in October could be causing an underestimation of SIC by CS2, as thin ice can be misclassified due to noise in the waveform returns and specular backscattering from leads can dominate radar echoes (Armitage and Davison, 2014; Wang et al., 2022). Although a correction has been applied to the lead and floe returns from CS2, this could

still be contributing to an underestimation of floes by CS2. Similarly, PMV retrievals have been found to be dependent on the ice composition within the footprint, underestimating ice concentration where thin ice dominates the footprint, but also overestimating thin ice when it represents a small proportion of the footprint (Shokr and Markus, 2006). These factors could all therefore be contributing to the offset in SIC between the datasets in October. As the CA is an area of low lead density (Tilling et al., 2019; Willmes et al., 2023) and thick sea ice

(Melling, 2022; Haas and Howell, 2015), these sensor challenges may be less than other areas of the Arctic.

The low correlation in February ($r = 0.1$ between CS2 and NT) and March ($r = < 0.2$ between CIS, NT and CS2) but low RMSD further suggests that challenging retrieval conditions could be generating variability between the datasets. In February and March, sea ice thicknesses and snow depths would generally be above the threshold for which thin ice would introduce uncertainties in sensor retrievals (Howell et al., 2016). However, significant

warming has been found across the CA in Spring (Tivy et al., 2011; Howell et al., 2016). As the NT algorithm uses static tie points, this introduces uncertainties in its retrievals, particularly for seasonal variations in accuracy and sensitivity to noise (Ivanova et al., 2015). Surface warming will also influence the contrast between different surface types; wet snow, for example, can appear as water in SAR backscatter. There has been limited uncertainty quantification of CIS ice chart accuracy under sub-optimal conditions and/or where SAR imagery does not have

clearly defined floes (Cheng et al., 2020).

The largest RMSD between CS2 and the other datasets tends to occur when SIC is high, between January and April. For CS2 and NT, the RMSD in these months is between 8% and 11%, and similarly between 7% and 10%



for CS2 and CIS. Atmospheric effects are known to impact PMV retrievals in areas of high ice concentration (Andersen et al., 2006; Ivanova et al., 2015) and ice variability is not always well-represented at high concentrations (Andersen et al., 2007). However, the NT algorithm can underestimate SIC in high concentrations (Andersen et al., 2007; Shokr and Markus, 2006), and the lower concentrations from CS2 at high SIC could therefore be an underestimate. CS2 has been found to identify floe waveform returns as ambiguous, and thus underestimate floe density in the Northwest Passage (Swiggs et al., 2024). Ambiguity over floes would thus lead to an overrepresentation of leads in the satellite footprint and an unrepresentative ratio of leads to floes. The nature of the bias adjustment used in this study means that floe densities cannot be adjusted when zero floe returns are identified in a grid cell. While this is unusual (See Fig. 2), we cannot rule out instances of 0% SIC due to CS2 falsely not identifying floes, rather than a true 0% SIC. Whilst we attempt to reduce these instances by removing grid cells that are 100% lead (Section 3), this could be contributing to the lower SIC estimate from CS2 and is thus a limitation of this method. This would particularly be a problem in high ice concentrations, where zero floe returns would have the greatest impact on overall SIC, and may explain the greater difference between CS2 and the other datasets.

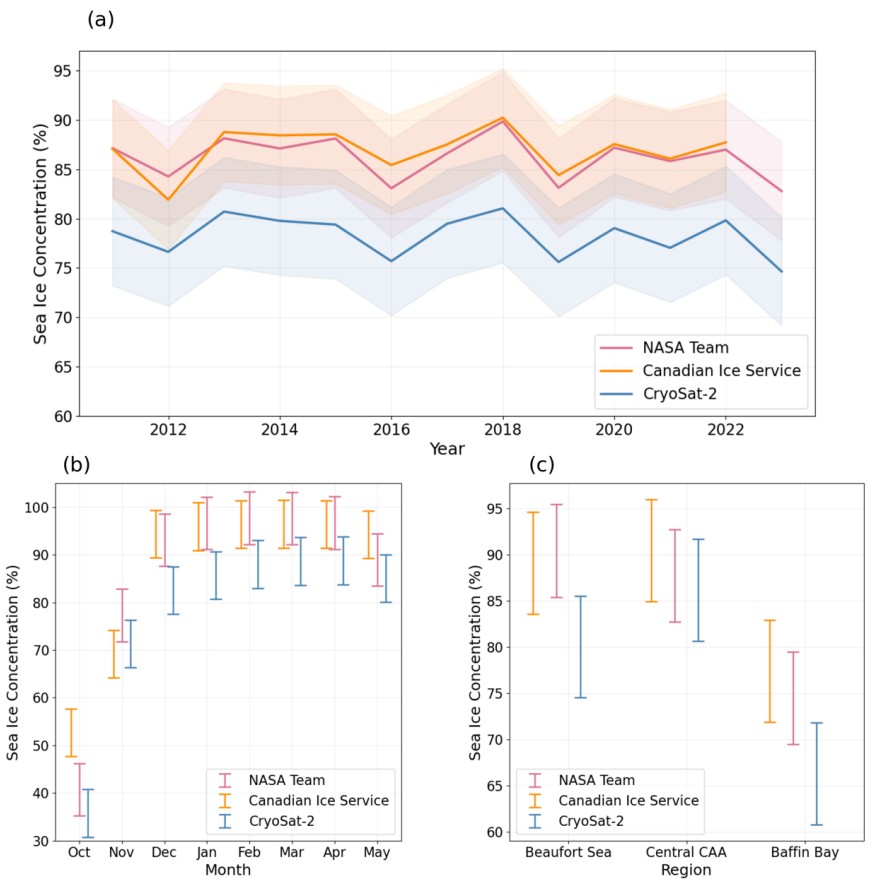



**Figure 4.** For NASA Team (pink), CryoSat-2 (blue) and the Canadian Ice Service (orange) in the Canadian Arctic (a) the average winter (October to May) from 2010 to 2023, (b) the monthly average sea ice concentration, (c) average sea ice concentration for specific study regions. A standard error metric is not given for Canadian Ice Service ice charts as the errors are nonuniform in space and time (Tivy et al., 2011), so we assume a standard of 5% either side of the mean as the concentration values are rounded to the nearest tenth. For NASA Team, the winter uncertainty value is assumed to be 5% (Digirolamo et al., 2022). For CryoSat-2, the uncertainty is 5.5% based on a Landsat 8 validation study (Swiggs et al., 2024).

### 4.3. Spatial Variations in Sea Ice Concentration

We also consider spatial differences in SIC determined from CS2, NT, and CIS. The datasets agree strongly in the Central CAA, with mean differences of just 1.6% between CS2 and NT, and 4.3% between CS2 and CIS (Fig. 5), where the sea ice is thickest and most landfast (Melling, 2022). The CAA is regarded as a challenging region for satellite observations due to the presence of complex ice types, landfast ice, land spillover contamination in PMV observations (Cavalieri et al., 1997) and low lead density inhibiting sea surface tie points in satellite altimetry (Tilling et al., 2019). However, these results suggest a good agreement between the datasets in this region, with little monthly variation in agreement, aside from in October and November for the reasons outlined above. In the CAA, the low lead density across landfast ice may also have improved the CS2 SIC estimate due to reduced instances of off-nadir ranging on CS2 waveform returns in this region. Furthermore, the location of the validation LS8 images used in the bias adjustment were largely centrally located over the CAA and may therefore have the greatest impact in this region (Swiggs et al., 2024).



**Figure 5. Maps showing the regional differences between the comparison datasets for each month, with red lines delineating the Beaufort Sea, Central Canadian Arctic Archipelago (CAA), and Baffin Bay. (Left) Average CryoSat-2 (CS2) sea ice concentration subtracted from NASA Team (NT) sea ice concentration. (Centre) Average CryoSat-2 sea ice concentration subtracted from Canadian Ice Service (CIS) sea ice concentration. (Right) Average Canadian Ice Service sea ice concentration subtracted from NASA Team sea ice concentration.**

In the Beaufort Sea, the differences between CS2 and the other datasets are more spatially and temporally uniform than in Baffin Bay (Fig. 5), suggesting a systematic offset between them. Increasing melt of MYI in the Beaufort



Sea (Babb et al., 2023) and concurrent increases in FYI (Tschudi et al., 2020) are likely generating challenging
ice conditions for satellite retrieval. Furthermore, the seasonal movement of the marginal ice zone in the Beaufort
Sea can be challenging to detect, particularly at monthly scales. In PMV observations, the low spatial resolution
and challenging atmospheric conditions over the marginal ice zone can introduce uncertainties (Andersen et al.,
2007; Meier, 2005), and the lower temporal sampling of CS2 and problems identifying thin ice in the marginal
ice zone are also likely contributing to differences in the Beaufort Sea.

The datasets have the greatest spatial variation in agreement in Baffin Bay; CS2 again observes the overall lowest
SIC in the region, but has a greater spatial agreement with NT. There are differences of up to 30% with CIS on
the persistently exposed ocean of the North Water Polynya in Nares Strait, which is subject to high volumes of
ice transport out of the Arctic Ocean into Baffin Bay (Melling, 2002; Howell et al., 2024). Here, the agreement
between CS2 and NT is best across the three datasets, suggesting that CIS could be overestimating SIC here. In
the lower latitudes of Baffin Bay, the high lead percentage and short floe length in the rest of Baffin Bay (Tilling
et al., 2019) could be causing CS2 to over detect leads and thus underestimate SIC. Further validation work should
be conducted in this region to ascertain if a further correction is required to the CS2 returns here.

**4.4. Evaluation of Sea Ice Concentration relative to Landsat 8**

We conduct a focused evaluation of the CS2 SIC product using high-resolution LS8 imagery. The overall
agreement between the SIC datasets at the LS8 sampling locations is high (Fig. 6 and Table B1) which is
encouraging for the CS2 sampling ability. LS8 and CS2 are better aligned at the middle ice concentrations (60 to
90%) compared to NT and CIS (Fig. 6c), and as such, CS2 and LS8 have the lowest RMSD of 6.5%, compared
to 7.8% for LS8 and NT, and 15.1% for LS8 and CIS. Similarly in the monthly observations, NT and CIS retrieve
a March SIC of 98% and 99%, respectively, whilst CS2 retrieves 94% and LS8 89% (Fig. 6a). There is also a
notable departure in agreement between CIS and LS8 is in October, where CIS records a high average SIC of
95.1%, compared to 79.5% for CS2, 77.8% for NT, and 66.5% for LS8. A focused view of the October average
SIC in the CAA shows that CS2 retrieves SIC between 0% and 30%, NT between 30% and 50%, and CIS between
50 and 70% in this region (Fig. 7). We evaluate the causes of these differences in the following paragraphs.



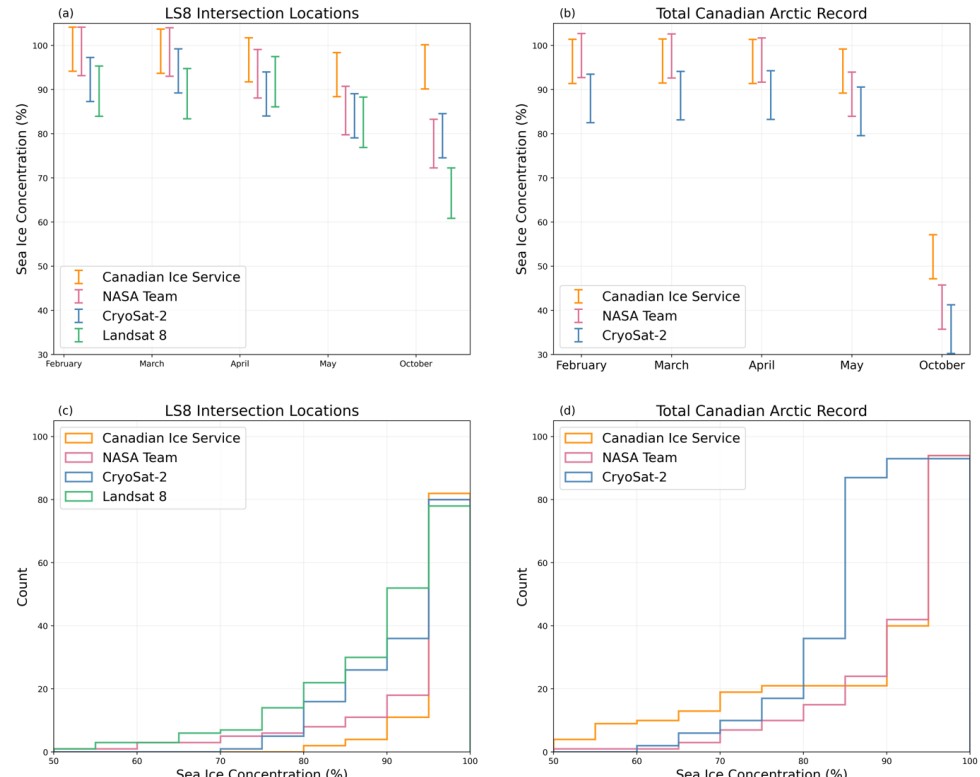

**Figure 6. A comparison between CryoSat-2 (blue), NASA Team (pink), Canadian Ice Service ice chart (orange) and Landsat 8 (green) sea ice concentrations. (a) Monthly sea ice concentration from the different sensors at the intersecting LS8 image locations. (b) Monthly average sea ice concentrations for the full record (only available from CryoSat-2, NASA Team, and the Canadian Ice Service). (c) Histograms of the monthly sea ice concentration from the different sensors at the intersecting LS8 image locations. (d) Histograms of the monthly average sea ice concentration for the full record (only available from CryoSat-2, NASA Team, and the Canadian Ice Service). The Landsat 8 error is based on sensitivity testing to the brightness temperature thresholds as outlined in Swiggs et al. (2024).**

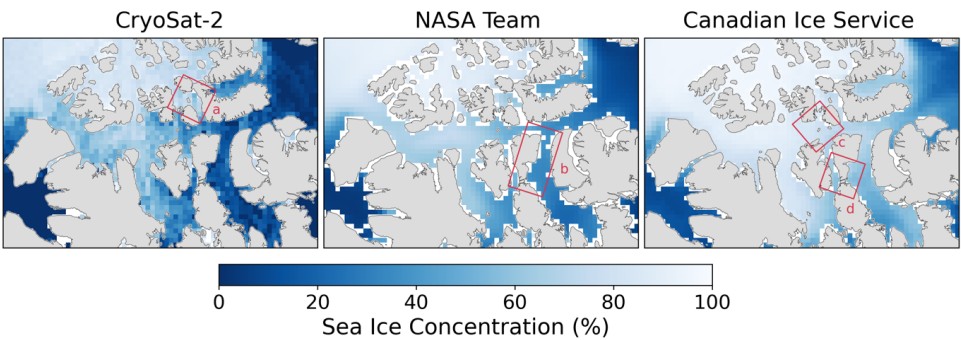

**Figure 7. A focused view of the October average sea ice concentration from CryoSat-2, NASA Team, and the Canadian Ice Service in the central Canadian Arctic Archipelago. The red boxes indicate the locations of panels a-d in Fig. 10.**



To assess the influence of temporal sampling on the agreement, we consider how the sea ice may have changed between the daily, weekly, and monthly datasets, and in particular, how representative the LS8 image is of the
overall monthly conditions. Temporal changes in sea ice conditions arise due to two main factors: dynamics and thermodynamics. We thus investigate temporal variation in ice drift (Fig. 8a), surface air temperature (Fig. 8b), and variation in SIC throughout each month using daily NT observations (Fig. 8c). Sea ice motion vectors are not available in the central CAA, although the sea ice here is largely landfast and immobile during winter (Tschudi et al., 2020; Howell et al., 2024). Peak mean drift of 6 km/day occurs in the Beaufort Sea and Baffin Bay in October
and January, respectively, when the ice pack is most mobile and during peak inflow and outflow from various regional flux gates (Bi et al., 2019). The regional averages in surface air temperature show peak lows of -28°C in February and highs of 4°C in Summer. The net effect of these factors is a seasonal variation in SIC that peaks during and after the melt season, and for Baffin Bay, in November due to movement of the ice edge. The minimum variation in SIC occurs during the peak SIC months of February and March. On average, the monthly variation in
SIC for October to May is 5.6%, which is smaller than the difference between the three datasets. Thus, whilst we consider temporal sampling in the differences between the datasets, we largely attribute the differences to other factors, namely their spatial sampling resolutions and treatment of refrozen leads, as we outline in the following paragraphs.

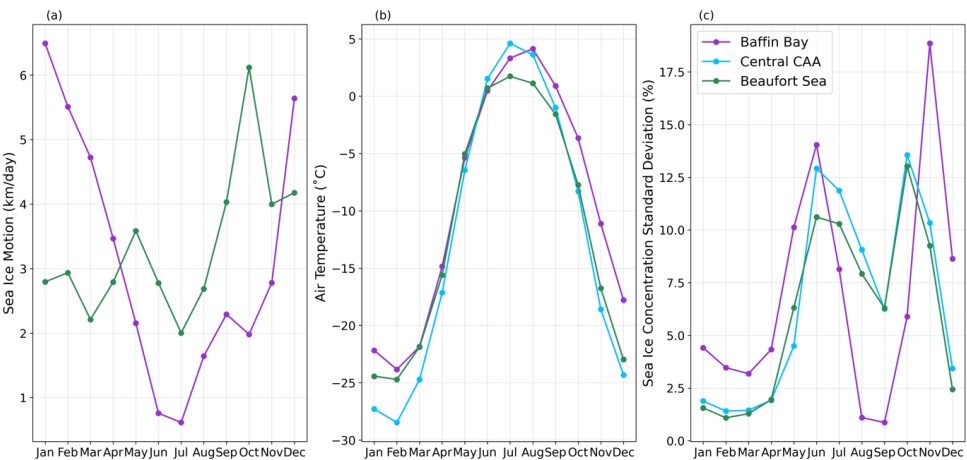

**Figure 8. (a) Regional monthly average sea ice motion calculated from calculated from averaged daily east and north components; data is not available in the central CAA (Tschudi et al., 2020). (b) Regional monthly average 2m surface air temperature from ERA5 reanalysis (Hersbach et al., 2023). (c) Regional monthly average sea ice concentration standard deviation calculated using daily NASA Team sea ice concentration values for each month.**

The CS2 SIC is generated from point measurements at approximately 0.3 km 1.5 km along- and across-track (Wingham et al., 2006). CS2 is therefore able to detect finer scale variations and leads in the ice pack, whereas the coarser resolution of PMV sensors means these are not always resolved (Laxon, 1994; Meier et al., 2015). CIS ice charts are generated from high-resolution Radarsat imagery, but as the SIC is averaged into tenths into polygons with a mean effective diameter of 45 km, some of the spatial variability within the polygons may
therefore be lost (Fig.1, Fig. B1). For LS8, which has an even finer scale spatial sampling resolution (between 30



and 100 m), a lower distribution of concentrations is observed; only 59% of observations are above 90% SIC, compared to 95% for CIS, 85% for NT, and 66% for CS2 at the intersecting locations. We therefore attribute the lower SIC from CS2 and LS8 to their higher lead detection. A large proportion of leads in the Arctic are less than 100 m wide (Kwok, 2010), and particularly in a region like the CAA which has large areas of immobile landfast

ice, the leads are likely often smaller than the spatial sampling resolution of PMV sensors. CS2 presents more variation in the sea ice pack, as its sensitivity to the presence of highly specular surfaces (i.e., leads) within the sensor footprint (Drinkwater, 1991) means it is able to detect even small leads, and our verification and bias adjustment of CS2 returns over the CAA has reduced the overestimation of leads from CS2. This suggests that the consistently lower SIC estimate from CS2 is due to its greater spatial sampling resolution that allows the

detection of small leads and icepack variation that can be missed by sensors.

In order to further understand differences between the datasets, we examine four images from the LS8 archive and the corresponding SIC from NT, CIS, and CS2 (Fig. 9). The selected images are from a variety of locations and months to show variable ice conditions. Across landfast sea ice (Fig. 9a), the gridded datasets exhibit similar high ice concentrations (>90%). In narrow land channels, the NT algorithm appears to be affected by land spillover

and thus has reduced coverage near the coastlines (Fig. 9a,c), whilst CIS demonstrates lower spatial variability due to averaging within the polygons (Fig. 9b,d). CS2 generally has broad spatial coverage and variation in SIC, but there are instances where it seems to underestimate SIC likely due to the presence of leads in the grid cell (Fig. 9b, d). Overall, CIS observes a higher SIC than other sensors, particularly in October (Fig. 9d), and the focused appraisal with the LS8 imagery suggests this may be due to how each sensor identifies refrozen leads. Refrozen

leads are challenging to map in thermal imagery due to their inconsistent brightness temperature, particularly when compared to the surrounding leads, making their identification somewhat subjective. In our classification scheme, we therefore choose to identify refrozen leads as leads (Fig. C1). The CIS ice charts, on the other hand, define a newly-frozen ice surface, including extremely thin frazil ice and slush, as ice (Canadian Ice Service, 2009). In CS2 processing, refrozen leads are most likely identified as leads, and the coarser spatial resolution of

NT suggest that these areas would also be identified as leads. These differing approaches to refrozen lead classification may therefore be causing differences between the datasets. For the four images in Fig. 9, we explore how classifying possible refrozen leads as ice would change the LS8 classification and find that LS8 image concentration would increase by 4.3% in Fig. 9b, 2.2% in Fig. 9c, 8.5% in Fig. 9d., and make no change in Fig. 9a (Fig. 10). Whilst not solely responsible for the differences, we thus find that the treatment of refrozen leads

can cause differences between the datasets in challenging retrieval conditions.







**Figure 9. Examples of four high-resolution Landsat 8 images masked into ocean and ice. Near-coincident sea ice concentration from NASA Team, Canadian Ice Service ice charts, and CryoSat-2 is shown in the following panels. Each row shows the NASA Team sea ice concentration from the day of the Landsat 8 image, the Canadian Ice Service ice chart from the week of the Landsat 8 image, and a 31-day average sea ice concentration from CryoSat-2, centred on the day of the Landsat 8 image (15 days +/- the day of retrieval). The corresponding Landsat 8 true colour images are in Fig. C1 and the influence of refrozen lead classification in Fig. 10.**

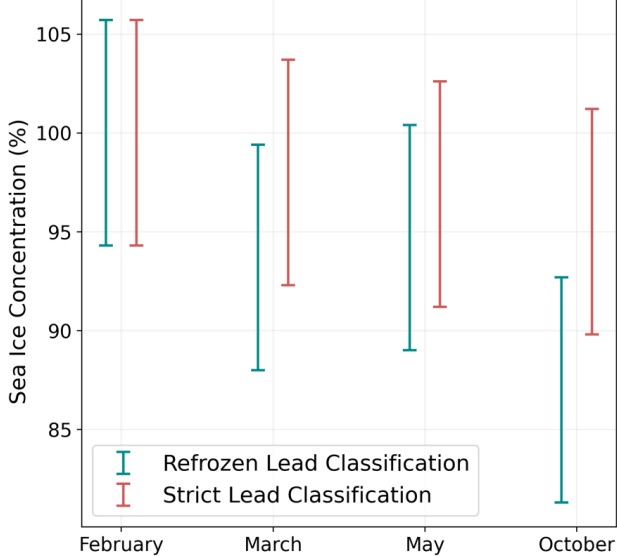

**Figure 10. For the four Landsat 8 images in Fig. 9, the differences in the Landsat 8 sea ice concentration when employing a strict lead classification versus the refrozen lead classification used for the full record.**

### 4.5. Potential and Limitations of Sea Ice Concentration from CryoSat-2

Our results present a CS2 SIC product in the CA and demonstrate its success in detecting fine scale sea ice pack variation. Comparisons to the NT algorithm, CIS ice charts, and LS8 imagery generally agree well, suggesting that CS2 can provide a complementary product in the CA to those that already exist. Currently, PMV observations of Arctic sea ice are at daily, year-round sampling resolution, and are therefore vital for Arctic-wide observations. CIS ice charts provide an essential continuous record of many sea ice parameters, but are not available in the wider Arctic. Whilst optical sensors such as LS8 are able to detect even finer scale variation, they are inhibited by cloud cover, and thus limited in their applicability to regional SIC estimates. Additional methods, such as SAR, require extensive computational power to classify and are therefore currently limited to regional and/or commercial use.

SIC from CS2 is aimed at complementing existing products, particularly as CS2 SIC requires PMV data to exclude ocean waveforms and is thus not wholly independent of PMV estimates (Section 3). This could be resolved in the future to create a standalone product from CS2; sea ice area with concentrations less than 50% represents less than 3% of the total sea ice area in the winter Arctic (Bocquet et al., 2024), and the application of a SIC threshold to diffuse waveforms was found to have a limited effect on overall waveform classification in the Northwest



Passage (Swiggs et al., 2024), suggesting that it could be removed entirely in certain regions and seasons. Fig. 2 also highlights the low ocean areas in winter. Furthermore, along-track altimetry data could be used to delineate the ice edge, thus negating the need for a separate dataset for defining ocean area. The use of PMV products in the CS2 SIC may also influence the overall agreement between the two datasets. However, as noted, the SIC threshold from PMV is limited in its application, particularly in a region like the CAA (Swiggs et al., 2024) and therefore has a narrow effect on the overall agreement.

Whilst it is not currently possible to produce an operational SIC estimate from CS2 in summer due to the challenges of melt pond formation on the surface of sea ice, CS2 can provide essential knowledge in sea ice pack dynamics throughout the rest of the year. This is particularly important in a region like the CA where complex ice processes will have significant implications for shipping and MYI ice export into the future. Furthermore, although SIC from CS2 has a lower temporal resolution than other SIC products, this does not limit its applicability for generating other commonly produced sea ice datasets, such as sea ice thickness and volume, which are also produced on monthly timescales. Currently, no observational sea ice thickness or volume estimate independent of PMV sensors exists. However, as highlighted in the application of a bias adjustment to CS2 lead and floe densities in this study, correcting for the challenges of off-nadir ranging and under-estimation of sea ice floes in CS2 processing, particularly in areas of low SIC and high lead density. This should be considered in Arctic-wide estimates of SIC from CS2.

## 5 Conclusions

SIC is a vital parameter for our understanding of sea ice dynamics and climate processes in the polar regions, and high-resolution estimates of SIC are of particular importance in the CA due to its significance for shipping, sea ice and freshwater transport, and MYI retention (Howell et al., 2024; Melling, 2022; Dawson et al., 2018). Currently, Arctic-wide estimates of SIC are from PMV sensors that have a high temporal resolution but low spatial resolution (e.g., Cavalieri et al., 1984; Comiso 1986; Tonboe et al., 2016). Although primarily derived from high-resolution SAR imagery, SIC in ice charts from CIS are rounded to the nearest tenth and may therefore lose some spatial variability (Canadian Ice Service, 2009). Satellite radar altimetry thus presents an opportunity for deriving SIC from higher resolution point measurements, thereby resolving finer-scale variability in the ice pack.

We utilise 13 years of observations from CS2 to produce a SIC estimate in the CA, correcting for the overestimations of leads and misclassification of floes in CS2 waveform returns with a bias adjustment generated from LS8 validation imagery (Swiggs et al., 2024). Our results show that CS2 can capture the expected seasonal cycle of SIC in the CA, with high SIC throughout Winter and Spring, and low concentrations in Autumn following the summer melt season. We find a good agreement between our results and SIC estimates from the NT algorithm and CIS ice charts, with correlations above r = 0.9 and RMSD values of 10% or lower for the monthly averages. A local comparison with LS8 imagery also demonstrates high agreement with the CS2 product, with an RMSD of just 3.3%. Whilst CS2 generally observes a lower SIC than CIS and NT, we largely attribute this to differences in spatial sampling and treatment of refrozen leads. Seasonal and regional variations in agreement are attributable to complex ice surfaces, high lead density, movement of the ice edge, and climatic conditions. Particularly in Baffin Bay, further work should be conducted to validate CS2 lead and floe density returns due to the region's high lead density and therefore potential underestimation of floes by CS2. Our results demonstrate the





complementarity of SIC estimates from CS2 to existing datasets, with CS2 able to capture fine-scale sea ice pack dynamics due to its high spatial sampling.

**Appendix A: Monthly Sea Ice Concentrations**

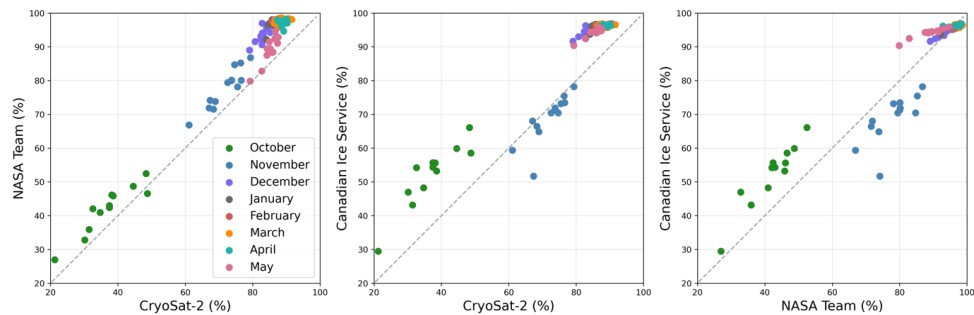

465

**Figure A1. Scatter plots of monthly average sea ice concentration values from CryoSat-2, NASA Team, and the Canadian Ice Service.**

**Table A1. Statistics (correlation (R)), root mean squared difference (RMSD) and absolute difference (Abs. Diff.) for all**
470 **monthly sea ice concentration values from CryoSat-2 (CS2), NASA Team (NT), and the Canadian Ice Service (CIS).**

|              | R    | RMSD | Abs Diff |
|--------------|------|------|----------|
| CS2 and NT   | 0.99 | 8.43 | 7.91     |
| CS2 and CIS  | 0.95 | 9.98 | 9.13     |
| NT and CIS   | 0.95 | 5.79 | 3.8      |



**Appendix B: Canadian Ice Service Charts**

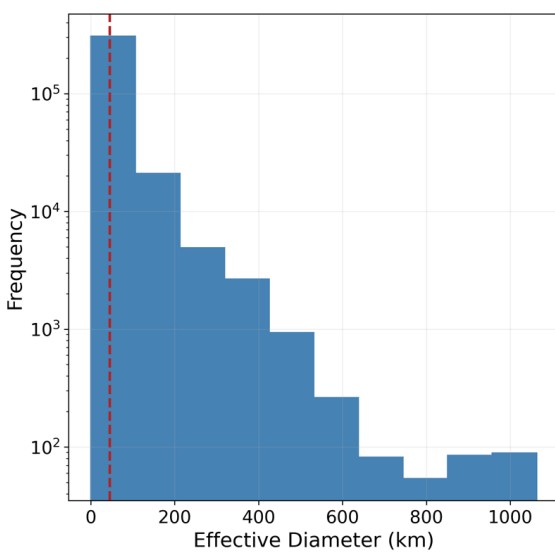

475 **Figure B1. Histogram showing the square root effective radius of areas of the Canadian Ice Service polygons used in the Western Arctic and Eastern Arctic from 2010 to 2023, on a logarithmic axis. The mean diameter of 45 km is shown with the red line.**

**Appendix C: Comparisons to Landsat 8**

**Table C1: Statistics for corresponding sea ice concentration values at the 82 Landsat 8 image locations, from CryoSat-**
480 **2 (CS2), NASA Team (NT), the Canadian Ice Service (CIS), and Landsat 8 (LS8). Correlation (R), root mean squared difference (RMSD), and absolute difference (Abs. Diff.).**

|  | R | RMSD | Abs Diff |
|---|---|---|---|
| **CS2 and LS8** | 0.7 | 3.29 | 3 |
| **CS2 and NT** | 0.98 | 4.51 | 4.12 |
| **CS2 and CIS** | 0.91 | 1.71 | 1.47 |
| **LS8 and NT** | 0.78 | 6.71 | 5.73 |
| **LS8 and CIS** | 0.93 | 2.1 | 1.9 |





| CIS and NT | 0.96 | 5 | 4.1 |
|---|---|---|---|

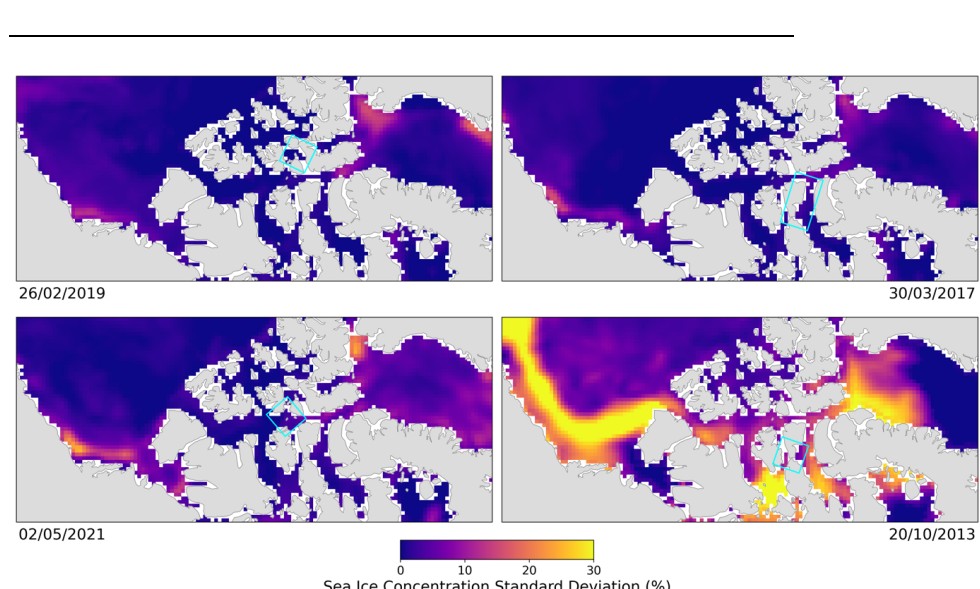

485 **Figure C1. For each of the comparison images used in Fig. 9., we show the standard deviation in sea ice concentration for 15 days +/- the date of the Landsat 8 image retrieval, calculated using daily NASA Team sea ice concentration data. The location of the Landsat 8 image is shown at the blue box.**




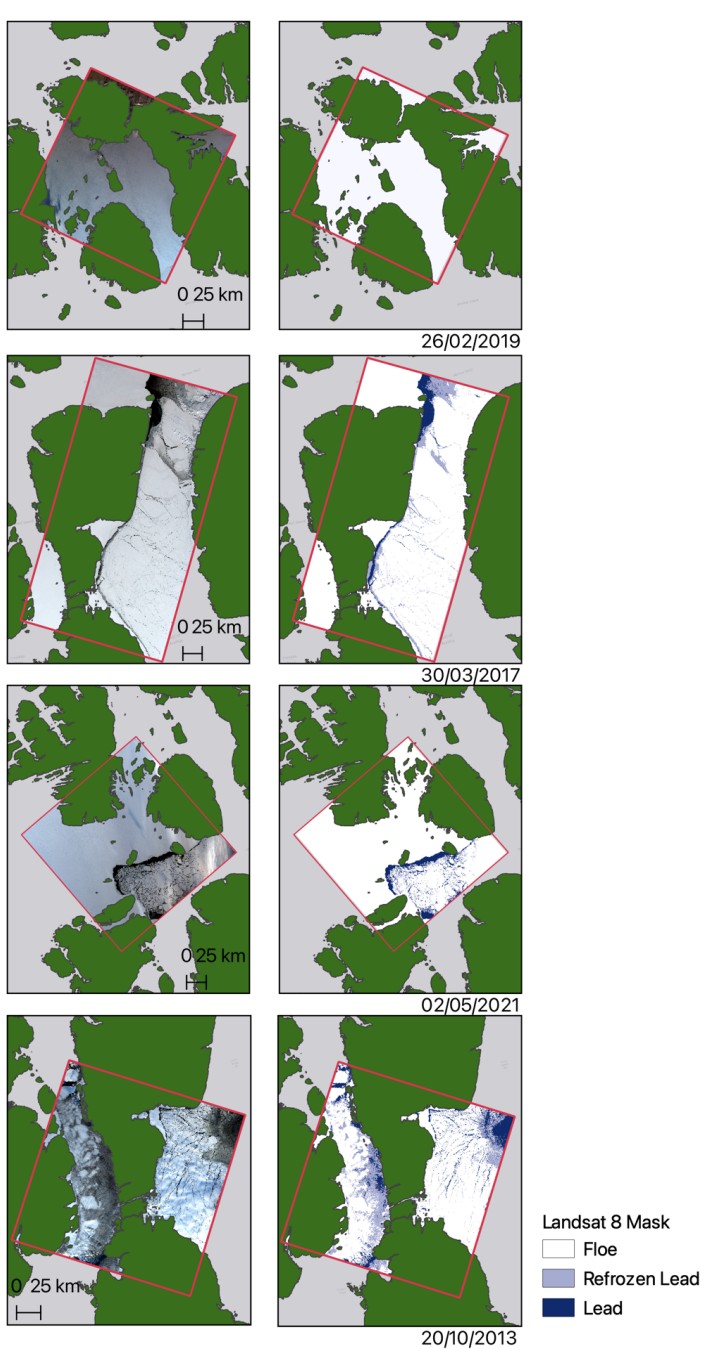

**Figure C2: The Landsat 8 True Colour images and corresponding masked ice and ocean used in Fig. 9. The images were masked using brightness temperature measurements from the Landsat 8 thermal infrared band and verified using true colour images (Section 2.4.)**



*Code and Data Availability.* The data used in this study are available as follows:

- L1b CryoSat-2 data are available for download from ESA via FTP (ftp://science-pds.cryosat.esa.int/)
- NASA Team data are available from the NSIDC (https://nsidc.org/data/nsidc-0051/versions/2)
- Canadian Ice Service sea ice concentrations are available from the NSIDC (https://nsidc.org/data/g02171/versions/1)
- Landsat 8 imagery are available from USGS via Google Earth Engine (https://developers.google.com/earth-engine/datasets/catalog/landsat-8/)
- ERA5 reanalysis data are available from the Copernicus Climate Data Store (https://cds.climate.copernicus.eu/datasets/reanalysis-era5-single-levels-monthly-means?tab=overview)
- Sea ice motion vectors are available from the NSIDC (https://nsidc.org/data/nsidc-0116/versions/4)

The code used to produce the figures in this study are available at https://github.com/amy-swiggs/Mapping_SeaIce_CAA_TC

*Author Contribution.* AES conducted the analysis, generated the figures, and wrote the manuscript. All the authors were involved in the study conceptualisation, scientific approach, and manuscript revisions.

*Competing Interests.* The authors declare that they have no conflict of interest.

*Acknowledgements.* The authors gratefully acknowledge the support of the European Space Agency in the preparation and completion of this study, including in hosting AES for a research visit at the Science Hub, ESRIN. The authors would also like to thank Jack Landy and Rachel Tilling for their feedback and helpful suggestions for this work. This work was supported by the Natural Environment Research Council (NERC) through the Centre for Polar Observation and Modelling and the DEFIANT project under Grant NE/W004747/1.

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
