# Peer review of "Mapping Sea Ice Concentration in the Canadian Arctic with CryoSat-2"

_EGUsphere, 2025_

## Referee Comment (RC2)

Review of
Mapping Sea Ice Concentration in the Canadian Arctic with CryoSat-2.

Authored by: Swiggs et al.,
The Cryosphere.
Manuscript Number: egusphere-2025-693

This paper presents a novel approach to estimating sea ice concentration in the Canadian Arctic from the along-track observations of CryoSat-2. Specifically, an existing classification of each waveform return into either a lead, floe, ocean or ambiguous point is gridded and summed over a month then used to calculate sea ice concentration (floe/[lead+floe+ocean]). An empirical correction is then applied based on previous work by the authors into issues with distinguishing leads and floes in CryoSat-2 returns. Results are computed for the full CryoSat-2 time-period (2010-2023) from October to May and compared broadly against sea ice concentration from the passive microwave record and regional ice charts. Further evaluation is done by comparing the CryoSat-2 results with 82 Landsat images. Overall, the agreement is good with the results highlighting the limitations of each method, which differ in spatial resolution, temporal resolution, and some specific nuance regarding refrozen leads. I find the paper to be well written and clear, with clear figures. Perhaps my biggest comment regards the regional focus, at first I thought the authors were specifically focused on the CAA where passive microwave concentration data is severely hampered by land, but the analysis covers the broader Canadian Arctic and includes both the Beaufort Sea and Baffin Bay where passive microwave data is adequate. I guess as a first proof of concept the regional focus is sufficient as it covers areas of both pack ice and seasonal landfast ice and areas where regional ice charts are readily available for another form of comparison. In saying this though, I would find a broader Arctic-wide product to be of interest.

Overall, my comments are quite minimal as the paper is well written and well presented, congratulations on this! I've listed some minor comments below that I think can help clarify and improve the paper, but nothing major.

- Line 79: most CS2 products end in April due to melt ponds, do you think there are issues with using CS2 from May or because you're only interested in the definition of floe vs water in the return this is suitable?

- Line 94: in brackets or at the end of the sentence comment on what they define as winter vs summer and how this compares to your seasonal focus? Does the error in PMV vary over your comparison (i.e. during May?).

- Line 110: I think it's worth clarifying that the "Western Canadian Arctic" and "Eastern Canadian Arctic" are regions defined by the CIS and not within this study.

- Line 197-199: I think this needs to be clarified. The Beaufort Gyre advects sea ice against the CAA/QEI, creating ridged ice. This ice is subsequently advected into the QEI where it maintains high SICs and largely becomes landfast throughout the study period. I'd suggest replacing "immobile" with "landfast" or including both terms here.

- Figure 1D: Is there a way to show the repeat coverage of CS-2? Is each track sampled roughly once per month or is there repeat? I think this gets at how episodic leads can be and the difference between consistent gridded sampling from PMV compared to infrequent sampling from along-track sensors like CS-2.

- Line 227: Is there a reference for the overestimation of SIC in charts? Also, I'd suggest editing to "... underestimation of SIC by CS2 and NT".

- Line 239: CA should be CAA.

- Line 241: I would have thought agreement would be higher in February as the CAA is landfast and the ice in the rest of the CA is relatively stable, with fewer leads. Can you elaborate on this?

- Line 244: The line about warming in spring stands out in the flow of the paragraph. I'd suggest moving it to the end of the paragraph as its worth noting but it doesn't immediately affect your results.

- Line 299: What are challenging conditions for sea ice retrievals? Does the change from MYI to FYI making it more difficult to interpret the data? Please consider revising or elaborating.

- Line 309: I think it's worth elaborating on why CIS shows high SIC compared to the other datasets around the North Water Polynya. I'd suggest its due to the ice charts showing concentrations of new and young ice in the polynya while other sensors may not see these thin ice types.

- Line 320: remove "is"

- Line 375: Suggest revising to "other sensors" or listing PMV and SAR.

- Line 430: Is it worth noting the work of Landy et al., (2022) who have developed a method for delineating floes during summer and thereby make the SIC algorithm possible?